# Understanding Event-Generation Networks via Uncertainties

**Marco Bellagente** [*1]   **Michel Luchmann** [*1]   **Manuel Haußmann** [2]   **Tilman Plehn** [1]

## Abstract

Generative models and normalizing flow-based models have made great progress in recent years, both in their theoretical development and in a growing number of applications. As such models become applied more and more, it increases the desire for predictive uncertainty to know when to trust the underlying model. In this extended abstract, we target the application area of Large Hadron Collider (LHC) simulations and show how to extend normalizing flows with probabilistic Bayesian Neural Network-based transformations to model LHC events with uncertainties.

## 1. Introduction

The role of first-principle simulations in our understanding of large data sets makes LHC physics stand out compared to many other areas of science. Three aspects define the application of modern big data methods in this field: (i) ATLAS and CMS deliver proper big data with excellent control over uncertainties, (ii) perturbative quantum field theory provides consistent precision predictions, and (iii) fast and reliable precision simulations generate events from first principles. The fact that experiments, field theory calculations, and simulations control their uncertainties implies that we can work with a complete uncertainty budget, including statistical, systematic, and theoretical uncertainties. To sustain this approach at the upcoming HL-LHC, with a data set more than 25 times the current Run 2 data set, the challenge is to provide faster simulations and to keep complete control of the uncertainties at the per-cent level and better.

In recent years it has been shown that modern machine learning and especially generative models can improve LHC event simulations in many ways (Butter & Plehn, 2020). See Section A in the appendix for an overview of recent work.

---

[*]Equal contribution  [1]Institut für Theoretische Physik, Universität Heidelberg, Germany [2]Heidelberg Collaboratory for Image Processing, Universität Heidelberg, Germany. Correspondence to: Tilman Plehn <plehn@uni-heidelberg.de>.

Third workshop on *Invertible Neural Networks, Normalizing Flows, and Explicit Likelihood Models* (ICML 2021). Copyright 2021 by the author(s).

The problem with these applications is that we know little about how these generative networks work and what the uncertainty on the generative network output is. As we will see, these two questions are closely related.

In general, we can track statistical and systematic uncertainties in neural network outputs with Bayesian neural networks (BNNs) (MacKay, 1995; Neal, 1995; Gal, 2016; Kendall & Gal, 2017). Such networks have been used in particle physics for a long time (Bhat & Prosper, 2005; Saucedo, 2007; Xu et al., 2008). For the LHC, they have been proposed to extract uncertainties in jet classification (Bollweg et al., 2020) and jet calibration (Kasieczka et al., 2020). They can cover essentially all uncertainties related to statistical, systematic, and structural limitations of the training sample (Nachman, 2020). Similar ideas can be used as part of ensemble techniques (Araz & Spannowsky, 2021).

We propose a *Bayesian invertible neural net (BINN)* which combines the flexibility of normalizing flow with BNNs and demonstrate it via a simple 2d toy example and finally, a semi-realistic LHC example. See the appendix for an extended discussion of these experiments and further results.

## 2. Generative Networks with Uncertainties

We start by reminding ourselves that we often assume that a generative model has learned a phase-space density perfectly. Hence, the only remaining source of uncertainty is the statistics of the generated sample binned in phase space. However, we know that such an assumption is not realistic (Bollweg et al., 2020; Kasieczka et al., 2020), and we need to estimate the effect of statistical or systematic limitations of the training data. The problem with such a statistical limitation is that it is turned into a systematic shortcoming of the generative model (Butter et al., 2019) — once we generate a new sample, the information on the training data is lost. The only way we might recover it is by training many networks and comparing their outcome. This is not a realistic or economical option for most applications, so we will show how an alternative solution could look.

**Invertible Neural Networks.** To model complex densities such as LHC phase-space distributions, we can employ normalizing flows (Rezende & Mohamed, 2015; Dinh et al., 2016; Kingma & Dhariwal, 2018; Kobyzev et al.,

2019). They use the fact we can transform a random variable $z \sim p_Z(z)$ using a bijective map $G : z \to x$ to a random variable $x = G(z)$ with the density

$$p_X(x) = p_Z(z) \left| \det \frac{\partial G(z)}{\partial z} \right|^{-1} = p_Z(\overline{G}(x)) \left| \det \frac{\partial \overline{G}(x)}{\partial x} \right| ,$$

where we defined $\overline{G} := G^{-1}$. Given a sample $z$ from the base distribution $p_Z$, we can use the map $G$ to generate a sample from the target distribution going in the forward direction and vice versa with a sample $x$ from the target.

For this to be a useful approach, we require the base distribution $p_Z$ to be simple enough to allow for effective sample generation, $G$ to be flexible enough for a non-trivial transformation, and its Jacobian determinant to be effectively computable. With these constraints, $G$ gives us a powerful generative pipeline to model the phase space density $p_X$. To fulfill them we choose the base distribution to be a multivariate Gaussian with mean zero and an identity matrix as the covariance, and rely on the real non-volume preserving flow (Dinh et al., 2016) in the invertible neural network (INN) formulation by Ardizzone et al. (2018) for $G$.

An INN composes multiple transformation maps into coupling layers with the following structure. The input vector $z$ into a layer is split in half, $z = (z_1, z_2)$, allowing us to compute the output $x = (x_1, x_2)$ of the layer as

$$\begin{pmatrix} x_1 \\ x_2 \end{pmatrix} = \begin{pmatrix} z_1 \odot e^{s_2(z_2)} + t_2(z_2) \\ z_2 \odot e^{s_1(x_1)} + t_1(x_1) \end{pmatrix} ,$$

where $s_i, t_i$ $(i = 1, 2)$ are small multi-layer perceptrons (MLP), and $\odot$ is the element-wise product. This structure allows both for easy invertibility as well as an easy Jacobian. Throughout, we refer to their weights jointly as $\theta$.

**Bayesian INN.** The invertible neural net provides us with a powerful generative model of the underlying data distribution. However, it lacks a mechanism to account for our uncertainty in the transformation parameters $\theta$ themselves. To model it, we switch from deterministic to probabilistic transformations, replacing the deterministic sub-networks $s_{1,2}$ and $t_{1,2}$ in each of the coupling layers with Bayesian neural nets. Placing priors over their weights $\theta \sim p(\theta)$ we get as the generative pipeline for our BINN

$$x|\theta \sim p_X(x|\theta) = p_Z(\overline{G}(x; \theta)) \left| \det \frac{\partial \overline{G}(x; \theta)}{\partial x} \right| .$$

Given our set of observations $\mathcal{D}$ we can rely on variational inference (Blei et al., 2017) to approximate the intractable posterior $p(\theta|\mathcal{D})$ with a mean-field Gaussian as the variational posterior $q_\phi(\theta)$. Learning then consists of maximizing the evidence lower bound (ELBO)

$$\mathcal{L} = \sum_{n=1}^{N} \mathbb{E}_{q_\phi(\theta)} \left[ \log p_X(x_n|\theta) \right] - \mathrm{KL}\big(q_\phi(\theta), p(\theta)\big)$$

$$= \sum_{n=1}^{N} \mathbb{E}_{q_\phi(\theta)} \left[ \log p_Z\big(\overline{G}(x_n; \theta)\big) + \log \left| \det \frac{\partial \overline{G}(x_n; \theta)}{\partial x_n} \right| \right]$$
$$- \mathrm{KL}\big(q_\phi(\theta), p(\theta)\big) ,$$

via stochastic gradient descent on the parameters $\phi$. By design all three terms, the log likelihood, log determinant, and the Kullback-Leibler (KL) divergence can be computed easily, and we can approximate the sum and the expectation with a minibatch and weight samples respectively.

## 3. Experiments

Before we tackle a semi-realistic LHC setup, we first study the behavior of BINNs for a set of toy examples, namely distributions over the minimally allowed two-dimensional parameter space where in one dimension the density is flat. Aside from the fact that these toy examples illustrate that the BINN actually constructs a meaningful uncertainty distribution, we will use the combination of density and uncertainty maps to analyse how an INN actually learns a density distributions. We will see that the INN describes the density map in the sense of a few-parameter fit, rather than numerically encoding patches over the parameter space independently. We discuss one of the three toy experiments here and refer the reader for the other two to the appendix. These low-dimensional examples allow us to visualize what is learned. However, the BINN also scales to higher-dimensional latent spaces as we will demonstrate on the MNIST data in the appendix as well. We there also provide details on the architecture and hyperparameters.

### 3.1. Toy Events with Uncertainties: The Wedge Ramp

Our first toy example is a two-dimensional ramp distribution, linear in one direction and flat in the other,

$$p(x, y) = \mathrm{Linear}(x \in [0, 1]) \cdot \mathrm{Const}(y \in [0, 1]) = x \cdot 2 .$$

The second factor ensures that the distribution $p(x, y)$ is normalized to one, and the network output is shown in Fig. 1 (upper). The output are unweighted events in the two-dimensional parameters space, $(x, y)$. We show one-dimensional distributions after marginalizing over the unobserved direction and find that the network reproduces the equation well. In the bottom row we include the predictive uncertainty given by the BINN. For this purpose we train a network on the two-dimensional parameter space and evaluate it for a set of points with $x \in [0, 1]$ and a constant $y$-value. In the left panel we indicate the predictive uncertainty as an error bar around the density estimate. Throughout the paper we always remove the phase space boundaries, because we know that the network is unstable there, and the uncertainties explode just like we expect. The relative uncertainty grows for small values of $x$ and hence small values of $p(x, y)$, and it covers the deviation of the

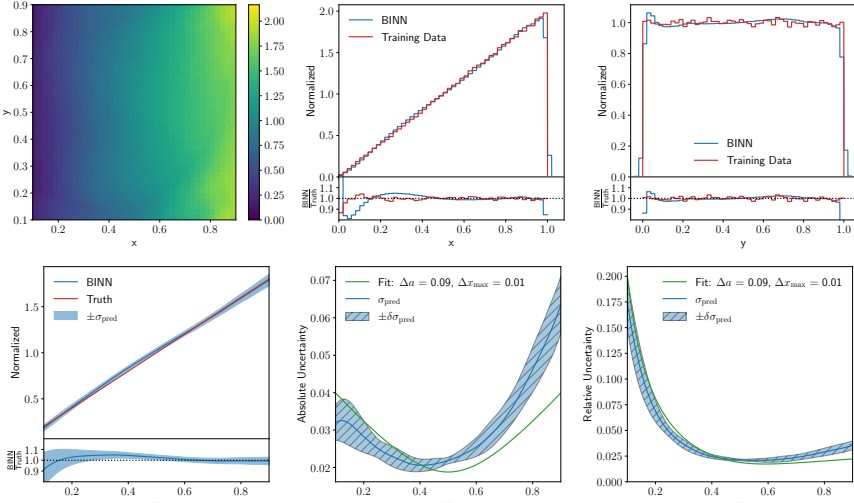

*Figure 1. (upper)* Two-dimensional and marginal densities for the linear wedge ramp. *(lower)* Density and predictive uncertainty distribution for the wedge ramp. In the left panel the density and uncertainty are averaged over several lines with constant $y$. In the central and right panels, the uncertainty band on $\sigma_{\text{pred}}$ is given by their variation. The green curve represents a two-parameter fit to (2).

extracted density from the true density well. These features are common to all our experiments. In the central and right panel of Fig. 1 we show the relative and absolute predictive uncertainties. The error bar indicates how much $\sigma_{\text{pred}}$ varies for different choices of $y$. We compute it as the standard deviation of different values of $\sigma_{\text{pred}}$, after confirming that the central values agree within this range. As expected, the relative uncertainty decreases towards larger $x$. However, the absolute uncertainty shows a distinctive minimum in $\sigma_{\text{pred}}$ around $x \approx 0.45$. This minimum is a common feature in all our training rounds, so we need to explain it.

To understand this non-trivial uncertainty distribution $\sigma_{\text{pred}}(x)$ we focus on the non-trivial $x$-coordinate and its linear behavior $p(x) = ax + b$ with $x \in [0, 1]$. As the model learns a density, we can remove $b$ by fixing the normalization, $p(x) = a(x - 0.5) + 1$. If we now assume that a network acts like a fit of $a$, as it will turn out useful, we can relate the uncertainty $\Delta a$ to an uncertainty in the density,

$$\sigma_{\text{pred}} \equiv \Delta p \approx |x - 0.5| \; \Delta a \; . \tag{1}$$

The absolute value appears because the uncertainties are defined to be positive, as encoded in the usual quadratic error propagation. The uncertainty distribution has a minimum at $x = 1/2$, close to the observed value in Fig. 1.

The differences between the simple prediction in (1) and our numerical findings in Fig. 1 are that the predictive uncertainty is not symmetric and does not reach zero. To account for these effects we can expand our very simple ansatz to $p(x) = ax + b$ with $x \in [x_{\text{min}}, x_{\text{max}}]$. Using the

normalization condition again we find

$$p(x) = ax + \frac{1 - \frac{a}{2}\left(x_{\text{max}}^2 - x_{\text{min}}^2\right)}{x_{\text{max}} - x_{\text{min}}} \; .$$

Again assuming a fit-like behavior of the flow network we expect for the predictive uncertainty

$$\sigma_{\text{pred}}^2 \equiv (\Delta p)^2 = \left(x - \frac{1}{2}\right)^2 (\Delta a)^2 + \left(1 + \frac{a}{2}\right)^2 (\Delta x_{\text{max}})^2$$
$$+ \left(1 - \frac{a}{2}\right)^2 (\Delta x_{\text{min}})^2 \; . \tag{2}$$

Adding $x_{\text{max}}$ adds an $x$-independent offset. Also accounting for $x_{\text{min}}$ does not change the $x$-dependence of predictive uncertainty. The slight shift of the minimum and the asymmetry between the lower and upper boundaries in $x$ are not explained by this argument. We ascribe them to boundary effects, specifically the challenge for the network to describe the correct approach towards $p(x) \to 0$.

The green line in the lower panels of Fig. 1 gives a two-parameter fit of $\Delta a$ and $\Delta x_{\text{max}}$ to the $\sigma_{\text{pred}}$ distribution from the BINN. It indicates that there is a hierarchy in the way the network extracts the $x$-independent term with high precision, whereas the uncertainty on the slope $a$ is around 4%.

### 3.2. LHC Events with Uncertainties

As a physics example we consider the Drell-Yan process

$$pp \to Z \to e^+ e^- \; ,$$

with its simple $2 \to 2$ phase space combined with the parton density. The training set consists of an unweighted set of

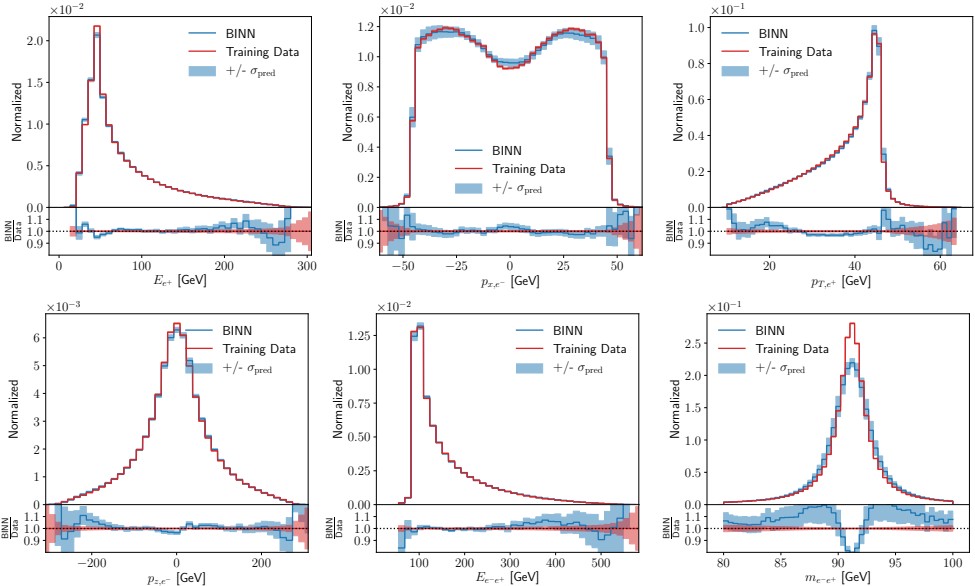

*Figure 2.* Marginalized kinematic distributions for the Drell-Yan process. We show the central prediction from the BINN and include the predictive uncertainty as the blue band. The red band indicates the statistical uncertainty of the training data per bin in the Gaussian limit.

4-vectors simulated with MADGRAPH5 (Alwall et al., 2014) at 13 TeV collider energy with the NNPDF2.3 parton densities (Ball et al., 2013). We fix the masses of the final-state leptons and enforce momentum conservation in the transverse direction, which leaves us with a four-dimensional phase space. In our discussion we limit ourselves to a sufficiently large set of one-dimensional distributions. For these marginalized uncertainties we follow the procedure laid out in Sec. C.1.4 with 50 samples in the BINN-weight space.

To start with, we show a set of generated kinematic distributions in Fig. 2. The positron energy features the expected strong peak from the $Z$-resonance. Its sizeable tail to larger energies is well described by the training data to $E_e \approx 280$ GeV. The central value learned by the BINN becomes unstable at slightly lower values of 250 GeV, as expected. The momentum component $p_x$ is not observable given the azimuthal symmetry of the detector, but it's broad distribution is nevertheless reproduced correctly. The predictive uncertainty covers the slight deviations over the entire range. What is observable at the LHC is the transverse momentum of the outgoing leptons, with a similar distribution as the energy, just with the $Z$-mass peak at the upper end of the distribution. Again, the predictive uncertainty determined by the BINN covers the slight deviations from the truth on the pole and in both tails. In the second row we show the $p_z$ component as an example for a strongly peaked distribution, similar to the Gaussian toy model in Sec. C.1.2.

While the energy of the lepton pair has a similar basic form as the individual energies, we also show the invariant mass of the electron-positron pair, which is described by the usual

Breit-Wigner peak. It is well known that this intermediate resonance is especially hard to learn for a network, because it forms a narrow, highly correlated phase space structure. Going beyond the precision shown here would for instance require an additional MMD loss (as e.g. in Butter et al., 2019; Bellagente et al., 2020b). This resonance peak is the only distribution, where the predictive uncertainty does not cover the deviation of the BINN density from the truth. This apparent failure corresponds to the fact that generative networks always overestimate the width and hence underestimate the height of this mass peak (Butter et al., 2019). This is an example of the network being limited by the expressive power in phase space resolution, generating an uncertainty which the Bayesian version cannot account for. See Sec. C.2.2 for further results.

## 4. Conclusion

Controlling the output of generative networks and quantifying their uncertainties is the main task for any application in LHC physics, be it in forward generation, inversion, or inference. We have proposed to use a Bayesian invertible neural network (BINN) to quantify the uncertainties from the network training for each generated event. For a series of two-dimensional toy models and an LHC-inspired application we have shown how the Bayesian setup indeed generates an uncertainty distribution, over the full phase space and over marginalized phase spaces. As expected, the learned uncertainty shrinks with an improved training statistics. Our method can be trivially extended from unweighted to weighted events by adapting the simple loss.

## Acknowledgements

We are very grateful for many discussions with Lynton Ardizzone, Anja Butter, Gregor Kasieczka, Ullrich Köthe, and Ramon Winterhalder. The research of TP is supported by the Deutsche Forschungsgemeinschaft (DFG, German Research Foundation) under grant 396021762 – TRR 257 *Particle Physics Phenomenology after the Higgs Discovery*. MB is supported by the International Max Planck School *Precision Tests of Fundamental Symmetries*. ML is supported by the DFG Research Training Group GK-1940, *Particle Physics Beyond the Standard Model*.

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

# APPENDIX

## A. Related Works on Generative Models for LHC event simulations

Promising techniques include generative adversarial networks (GAN) (Goodfellow et al., 2014; Creswell et al., 2018; Butter et al., 2020a), variational autoencoders (Kingma & Welling, 2014; 2019), and normalizing flows (Rezende & Mohamed, 2015; Kobyzev et al., 2020; Papamakarios et al., 2019; Kobyzev et al., 2019; Müller et al., 2018), including invertible networks (INNs) (Ardizzone et al., 2018; Dinh et al., 2016; Kingma & Dhariwal, 2018). They can improve phase space integration (Klimek & Perelstein, 2018; Chen et al., 2021), phase space sampling (Bothmann et al., 2020; Gao et al., 2020b;a), and amplitude computations (Bishara & Montull, 2019; Badger & Bullock, 2020). Further developments are fully NN-based event generation (Otten et al., 2019; Hashemi et al., 2019; Di Sipio et al., 2020; Butter et al., 2019; Alanazi et al., 2020), event subtraction (Butter et al., 2020b), event unweighting (Stienen & Verheyen, 2021; Backes et al., 2020), detector simulation (Paganini et al., 2018a;b; Musella & Pandolfi, 2018; Erdmann et al., 2018; 2019; ATLAS Collaboration, 2018; 2019; Belayneh et al., 2020; Buhmann et al., 2020; 2021), or parton showering (Bothmann & Debbio, 2019; de Oliveira et al., 2017; Monk, 2018; Andreassen et al., 2019; Dohi, 2020). Generative models will also improve searches for physics beyond the Standard Model (Lin et al., 2019), anomaly detection (Nachman & Shih, 2020; Knapp et al., 2020), detector resolution (Di Bello et al., 2021; Baldi et al., 2020), and inference (Brehmer & Cranmer, 2020; Radev et al., 2020; Bieringer et al., 2020). Finally, conditional GANs and INNs allow us to invert the simulation chain to unfold detector effects (Datta et al., 2018; Bellagente et al., 2020b) and extract the hard scattering process at parton level (Bellagente et al., 2020a).

## B. Uncertainties on Event Samples

Uncertainties on a simulated kinematic or phase space distribution are crucial for any LHC analysis. For instance, we need to know to what degree we can trust a simulated $p_T$-distribution in mono-jet search for dark matter. We denote the complete phase space weight for a given phase space point as $p(x)$, and can then illustrate a total cross section as

$$\sigma_{\text{tot}} = \int_0^1 dx \, p(x) \quad \text{with} \quad p(x) > 0 \, . \qquad (3)$$

In this simplified notation $x$ stands for a generally multi-dimensional phase space. For each phase space position, we can also define an uncertainty $\sigma(x)$.

One contribution to the error budget are systematic and theory uncertainties, $\sigma_{\text{th/sys}}(x)$. The former reflect our ignorance of aspects of the training data, which do not decrease when we increase the amount of training data. The latter captures the degree to which we trust our prediction, for instance based on self-consistency arguments. For example accounting for large, momentum-dependent logarithms we can compute it from the phase space position, or for an unweighted event, alone. If we use a numerical variation of the factorization and renormalization scale to estimate a theory uncertainty, we typically re-weight events with the scales. Another uncertainty arises from the statistical limitations of the training data, $\sigma_{\text{stat}}(x)$. For instance in mono-jet production, the tails of the predicted $p_T$-distribution for the Standard Model will at some point be statistics limited. In the Gaussian limit, a statistical uncertainty can be defined by binning the phase space and in that limit we expect a scaling like $\sigma_{\text{stat}}(x) \sim \sqrt{p(x)}$, and we will test that hypothesis in detail in Sec. C.1.

Once we know the uncertainties as a function of the phase space position, we can account for them as additional entries in unweighted or weighted events. For instance, relative uncertainties can be easily added to unweighted events,

$$\text{ev}_i = \begin{pmatrix} \sigma_{\text{stat}}/p \\ \sigma_{\text{syst}}/p \\ \sigma_{\text{th}}/p \\ \{x_{\mu,j}\} \\ \{p_{\mu,j}\} \end{pmatrix} \, , \quad \text{with } \mu = 0 \dots 3 \text{ for each particle } j.$$

The entries $\sigma$ or $\sigma/p$ are smooth functions of phase space. The challenge in working with this definition is how to extract $\sigma_{\text{stat}}$ without binning. We will show how Bayesian networks give us access to limited information in the training data. Specific theory and systematics counterparts can be either computed directly or extracted by appropriately modifying the training data (Bollweg et al., 2020; Kasieczka et al., 2020).

## C. Further Experiments

### C.1. Toy Events with Uncertainties

The default architecture for our toy models is a network with 32 units per layer, three layers per coupling block, and a total of 20 coupling blocks. It's implemented in PYTORCH (Paszke et al., 2019) relying partially on the FreIA library[1] More details are given in Tab. 1. The most relevant hyperparameter is the number of coupling blocks in that more blocks provide a more stable performance with respect to several trainings of the same architecture. Generally, moderate changes for instance of the number of units per layer do not have a visible impact on the performance.

---

[1]Framework for Easily Invertible Architectures, https://github.com/VLL-HD/FrEIA

*Table 1.* Hyper-parameters for all toy models, implemented in PY-TORCH (v1.4.0) (Paszke et al., 2019).

| Parameter | Flow |
|---|---|
| Hidden layers (per block) | 3 |
| Units per hidden layer | 32 |
| Batch size | 512 |
| Epochs | 300 |
| Trainable weights | 75k |
| Optimizer | Adam |
| $(\alpha, \beta_1, \beta_2)$ | $(1 \times 10^{-3}, 0.9, 0.999)$ |
| Coupling layers | 20 |
| Training size | 300k |
| Prior width | 1 |

For each of the trainings we use a sample of 300k events. The widths of the Gaussian priors are set to one. We check that variations of this over several orders of magnitude did not have a significant impact on the performance.

### C.1.1. KICKER RAMP

We can test our findings from the linear wedge ramp using the slightly more complex quadratic or kicker ramp,

$$p(x,y) = \text{Quadr}(x \in [0,1]) \times \text{Const}(y \in [0,1]) = x^2 \times 3 .$$

We show the results from the network training for the density in Fig. 3 and find that the network describes the density well, limited largely by the flat, low-statistics approach towards the lower boundary with $p(x) \to 0$.

In complete analogy to Fig. 1 we show the complete BINN output with the density $p(x,y)$ and the predictive uncertainty $\sigma_{\text{pred}}(x,y)$ in Fig. 4. As for the linear case, the BINN reproduces the density well, deviations from the truth being within the predictive uncertainty in all points of phase space. We remove the phase space boundaries, where the network becomes unstable and the predictive uncertainties grows correspondingly. The indicated error bar on $\sigma_{\text{pred}}(x,y)$ is given by the variation of the predictions for different $y$-values, after ensuring that their central values agree. The relative uncertainty at the lower boundary $x = 0$ is large, reflecting the statistical limitation of this phase-space region. An interesting feature appears again in the absolute uncertainty, namely a maximum-minimum combination as a function of $x$.

Again in analogy to the wedge ramp, we start with the parametrization of the density

$$p(x) = a\,(x - x_0)^2 \qquad \text{with} \qquad x \in [x_0, x_{\max}] , \quad (4)$$

where we assume that the lower boundary coincides with the minimum and there is no constant offset. We choose to describe this density through the minimum position $x_0$,

coinciding the the lower end of the $x$-range, and $x_{\max}$ as the second parameter. The parameter $a$ can be eliminated through the normalization condition and we find

$$p(x) = 3\frac{(x - x_0)^2}{(x_{\max} - x_0)^3} . \qquad (5)$$

If we vary $x_0$ and $x_{\max}$ we can trace two contributions to the uncertainty in the density,

$$\sigma_{\text{pred}} \equiv \Delta p \supset \frac{9}{(x_{\max} - x_0)^4}$$
$$\cdot \left| (x - x_0) \left( x - \frac{x_0}{3} - \frac{2x_{\max}}{3} \right) \right| \Delta x_0$$

and

$$\sigma_{\text{pred}} \equiv \Delta p \supset \frac{9}{(x_{\max} - x_0)^4}\,(x - x_0)^2\,\Delta x_{\max} , \quad (6)$$

one from the variation of $x_0$ and one from the variation of $x_{\max}$. In analogy to (2) they need to be added in quadrature. If the uncertainty on $\Delta x_0$ dominates, the uncertainty has a trivial minimum at $x = 0$ and a non-trivial minimum at $x = 2/3$. From $\Delta x_{\max}$ we get another contribution which scales like $\Delta p \propto p(x)$. In Fig. 4 we clearly observe both contributions, and the green line in the lower panels is given by the corresponding 2-parameter fig to the $\sigma_{\text{pred}}$ distribution from the BINN.

### C.1.2. GAUSSIAN RING

Our third example is a two dimensional Gaussian ring, which in terms of polar coordinates reads

$$p(r, \phi) = \text{Gauss}(r > 0; \mu = 4, w = 1)$$
$$\times \text{Const}(\phi \in [0, \pi]) , \qquad (7)$$

We define the Gaussian density as the usual

$$\text{Gauss}(r) = \frac{1}{\sqrt{2\pi}\,w} \exp\left[ -\frac{1}{2w^2}(r - \mu)^2 \right] \qquad (8)$$

The density defined in (7) can be translated into Cartesian coordinates as

$$p(x,y) = \text{Gauss}(r(x,y); \mu = 4, w = 1)$$
$$\times \text{Const}(\phi(x,y) \in [0, \pi]) \times \frac{1}{r(x,y)} , \qquad (9)$$

where the additional factor $1/r$ comes from the Jacobian. We train the BINN on Cartesian coordinates, just like in the two examples before, and limit ourselves to $y > 0$ to avoid problems induced by learning a non-trivial topology in mapping the latent and phase spaces. In Fig. 5 we once again see that our network describes the true two-dimensional density well.

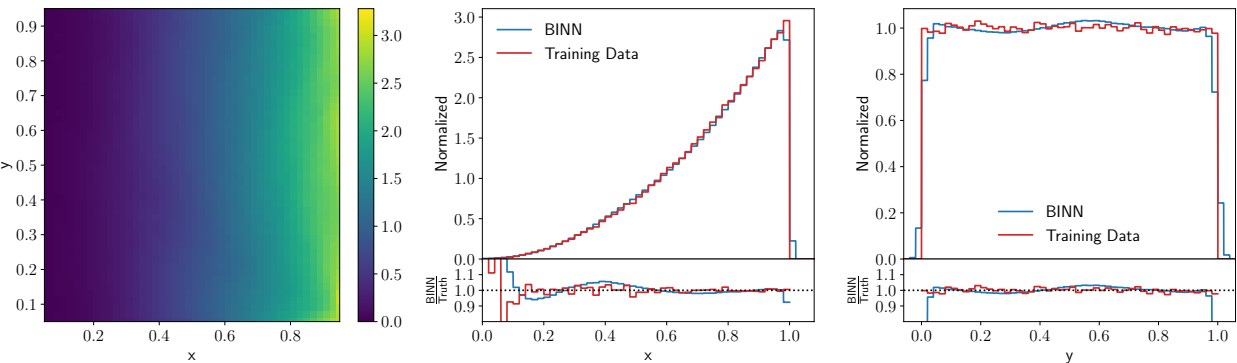

*Figure 3.* Two-dimensional and marginal densities for the quadratic kicker ramp.

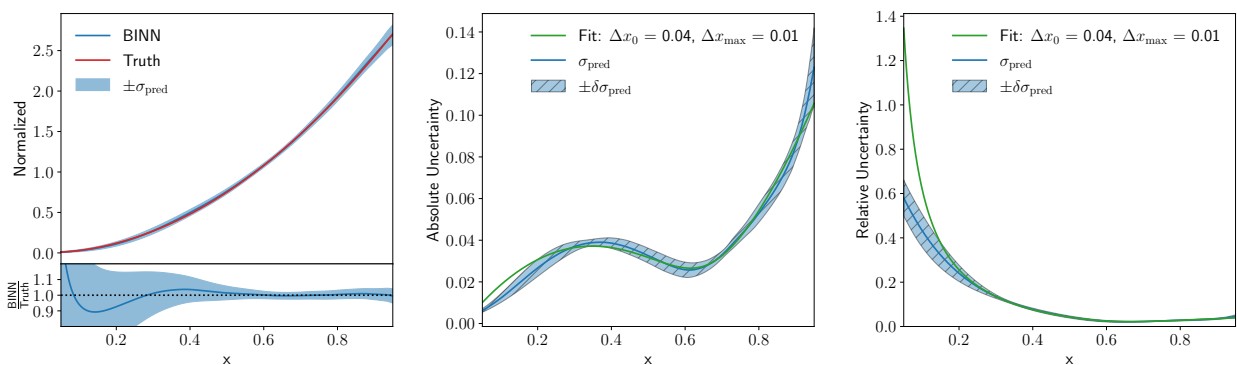

*Figure 4.* Density and predictive uncertainty distribution for the kicker ramp. In the left panel the density and uncertainty are averaged over several lines with constant $y$. In the central and right panels, the uncertainty band on $\sigma_{\text{pred}}$ is given by their variation. The green curve represents a two-parameter fit to (6).

In Fig. 6 we show the Cartesian density but evaluated on a line of constant angle. This form includes the Jacobian and has the expected, slightly shifted peak position at $r_{\max} = 2 + \sqrt{3} = 3.73$. The BINN returns a predictive uncertainty, which grows towards both boundaries. The error band easily covers the deviation of the density learned by the BINN and the true density. While the relative predictive uncertainty appears to have a simple minimum around the peak of the density, we again see that the absolute uncertainty has a distinct structure with a local minimum right at the peak. The question is what we can learn about the INN from this pattern in the BINN.

As before, we describe our distribution in the relevant direction in terms of convenient fit parameters. For the Gaussian radial density these are the mean $\mu$ and the width $w$ used in (7). The contributions driven by the extraction of the mean in Cartesian coordinates reads

$$\sigma_{\text{pred}} \equiv \Delta p \supset \left| \frac{G(r)}{r} \frac{\mu - r}{w^2} \right| \Delta \mu$$

and

$$\sigma_{\text{pred}} \equiv \Delta p \supset \left| \frac{(r - \mu)^2}{w^3} - \frac{1}{w} \right| \Delta w . \tag{10}$$

In analogy to (2) the two contributions need to be added in quadrature for the full, fit-like uncertainty. The contribution from the the mean has a minimum at $r = \mu = 4$ and is otherwise dominated by the exponential behavior of the Gaussian, just as we observe in the BINN result. In the opposite limit of $\Delta \mu \ll \Delta w$ the uncertainty develops the maxima at $r = 3$ and $r = 5$, which we observe in Fig. 6. In the lower panels we show a one-parameter fit of the BINN output and find that the network determined the mean of the Gaussian as $\mu = 4 \pm 0.037$.

### C.1.3. ERRORS VS TRAINING STATISTICS

Even though it is clear from the above discussion that we cannot expect the predictive uncertainties to have a simple scaling pattern, like for the regression (Kasieczka et al., 2020) and classification (Bollweg et al., 2020) networks, there still remains the question how the BINN uncertainties change with the size of the training sample.

In Fig. 7 we show how the BINN predictions for the density and uncertainty change if we vary the training sample size from 10k events to 1M training events. Note that for all

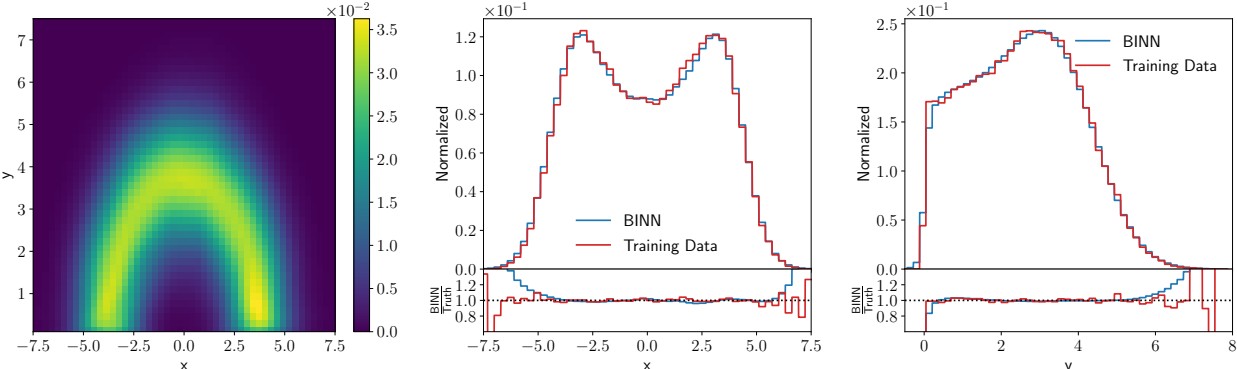

*Figure 5.* Two-dimensional and marginal densities for the Gaussian (half-)ring.

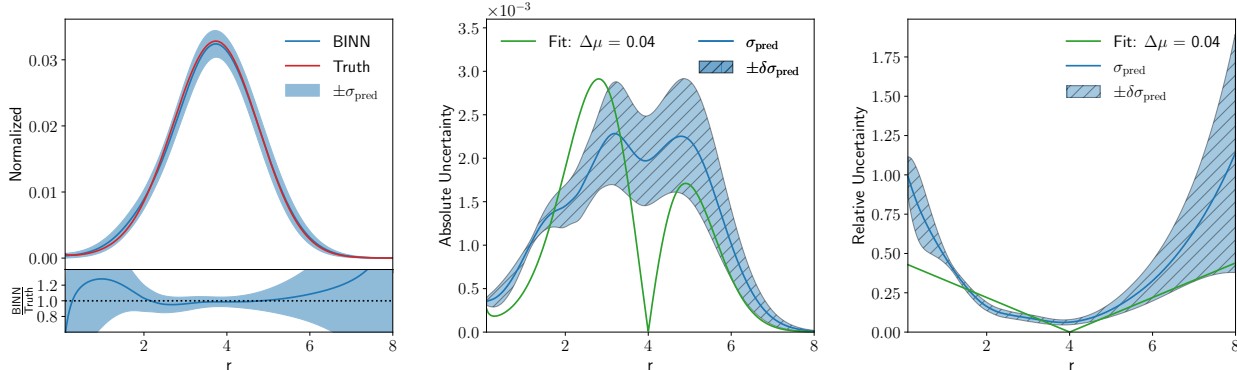

*Figure 6.* Cartesian density and predictive uncertainty distribution for the Gaussian ring. In the left panel the density and uncertainty are averaged over several lines with constant $\phi$. In the central and right panels, the uncertainty band on $\sigma_{\text{pred}}$ is given by their variation. The green curve represents a two-parameter fit to (10).

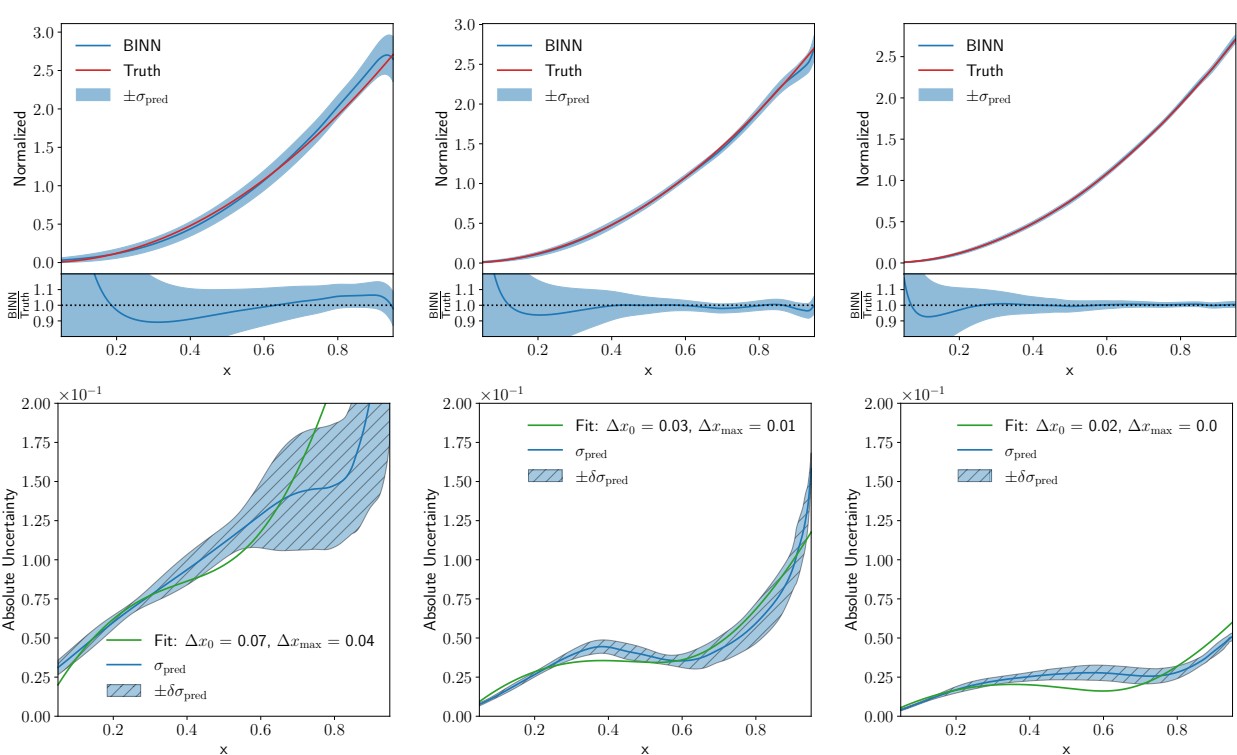

*Figure 7.* Dependence of the density (upper) and absolute uncertainty (lower) on the training statistics for the kicker ramp. We illustrate BINNs trained on 10k, 100k, and 1M events (left to right), to be compared to 300k events used for Fig. 4. Our training routine ensures that all models receive the same number of weights updates, regardless of the training set size.

toy models, including the kicker ramp in Sec. C.1.1, we use 300k training events. For the small 10k training sample, we see that the instability of the BINN density becomes visible even for our reduced $x$-range. The peak-dip pattern of the absolute uncertainty, characteristic for the kicker ramp, is also hardly visible, indicating that the network has not learned the density well enough to determine its shape. Finally, the variation of the predictive density explodes for $x > 0.4$, confirming the picture of a poorly trained BINN. As a rough estimate, the absolute uncertainty at $x = 0.5$ with a density value $p(x, y) = 0.75$ ranges around $\sigma_{\text{pred}} = 0.11 \ldots 0.15$.

For 100k training events we see that the patterns discussed in Sec. C.1.1 begin to form. The density and uncertainty encoded in the network are stable, and the peak-dip with a minimum around $x = 2/3$ becomes visible. As a rough estimate we can read off $\sigma_{\text{pred}}(0.5) \approx 0.06 \pm 0.03$. For 1M training events the picture improves even more and the network extracts a stable uncertainty of $\sigma_{\text{pred}}(0.5) \approx 0.03 \pm 0.01$. Crucially, the dip around $x \approx 2/3$ remains, and even compared to Fig. 4 with its 300k training events the density and uncertainty at the upper phase space boundary are much better controlled.

Finally, we briefly comment on a frequentist interpretation of the BINN output. We know from simpler Bayesian networks (Bollweg et al., 2020; Kasieczka et al., 2020) that it is possible to reproduce the predictive uncertainty using an ensemble of deterministic networks with the same architecture. However, from those studies we also know that our class of Bayesian networks has a very efficient built-in regularization, so this kind of comparison is not trivial. For the BINN results shown in this paper we find that the detailed patterns in the absolute uncertainties are extracted by the Bayesian network much more effectively than they would be for ensembles of deterministic INNs. For naive implementations with a similar network size and no fine-tuned regularization these patterns are somewhat harder to extract. On the other hand, in stable regions without distinctive patterns the spread of ensembles of deterministic networks reproduces the predictive uncertainty reported by the BINN.

### C.1.4. MARGINALIZING PHASE SPACE

Before we move to a more LHC-related problem, we need to study how the BINN provides uncertainties for marginalized kinematic distribution. In all three toy examples the two-dimensional phase space consists of one physical and one trivial direction. For instance, the kicker ramp in Sec. C.1.1 has a quadratic physical direction, and in a typical phase space problem we would integrate out the trivial, constant direction and show a one-dimensional kinematic distribution. From our effectively one-dimensional uncertainty extraction we know that the absolute uncertainty has a characteristic

maximum-minimum combination, as seen in the lower-right panel of Fig. 4.

To compute the uncertainty for a properly marginalized phase space direction, we remind ourselves how the BINN computes the density and the predictive uncertainty by sampling over the weights,

$$p(x, y) = \int d\theta \, q(\theta) \, p(x, y|\theta)$$

$$\sigma_{\text{pred}}^2(x, y) = \int d\theta \, q(\theta) \left[ p(x, y|\theta) - p(x, y) \right]^2 . \quad (11)$$

If we integrate over the $y$-direction, the marginalized density is defined as

$$
\begin{aligned}
p(x) &= \int dy \, p(x, y) \\
&= \int dy d\theta \, q(\theta) \, p(x, y|\theta) \\
&= \int d\theta \, q(\theta) \int dy \, p(x, y|\theta) \\
&\equiv \int d\theta \, q(\theta) \, p(x|\theta) , \quad (12)
\end{aligned}
$$

which implicitly defines $p(x|\theta)$ in the last step, notably without providing us with a way to extract it in a closed form. The key step in this definition is that we exchange the order of the $y$ and $\theta$ integrations. Nevertheless, with this definition at hand, we can *define* the uncertainty on the marginalized distribution as

$$\sigma_{\text{pred}}^2(x) = \int d\theta \, q(\theta) \left[ p(x|\theta) - p(x) \right]^2 . \quad (13)$$

We illustrate this construction with a trivial $p(x, y) = p(x, y_0)$, where we can replace the trivial $y$-dependence by a fixed choice $y = y_0$ just like for the wedge and kicker ramps. Here we find, modulo a normalization constant in the $y$-integration

$$
\begin{aligned}
\sigma_{\text{pred}}^2(x) &= \int d\theta \, q(\theta) \left[ p(x|\theta) - p(x) \right]^2 \\
&= \int d\theta \, q(\theta) \int dy \, \left[ p(x, y_0|\theta) - p(x, y_0) \right]^2 \\
&= \int dy d\theta \, q(\theta) \, \left[ p(x, y_0|\theta) - p(x, y_0) \right]^2 \\
&= \int dy \, \sigma_{\text{pred}}^2(x, y_0) = \sigma_{\text{pred}}^2(x, y_0) . \quad (14)
\end{aligned}
$$

Adding a trivial $y$-direction does not affect the predictive uncertainty in the physical $x$-direction.

As mentioned above, unlike for the joint density, $p(x, y|\theta)$ we do not know the closed form of the marginal distributions $p(x)$ or $p(x|\theta)$. Instead, we can approximate the marginalized uncertainties through a combined sampling in $y$ and

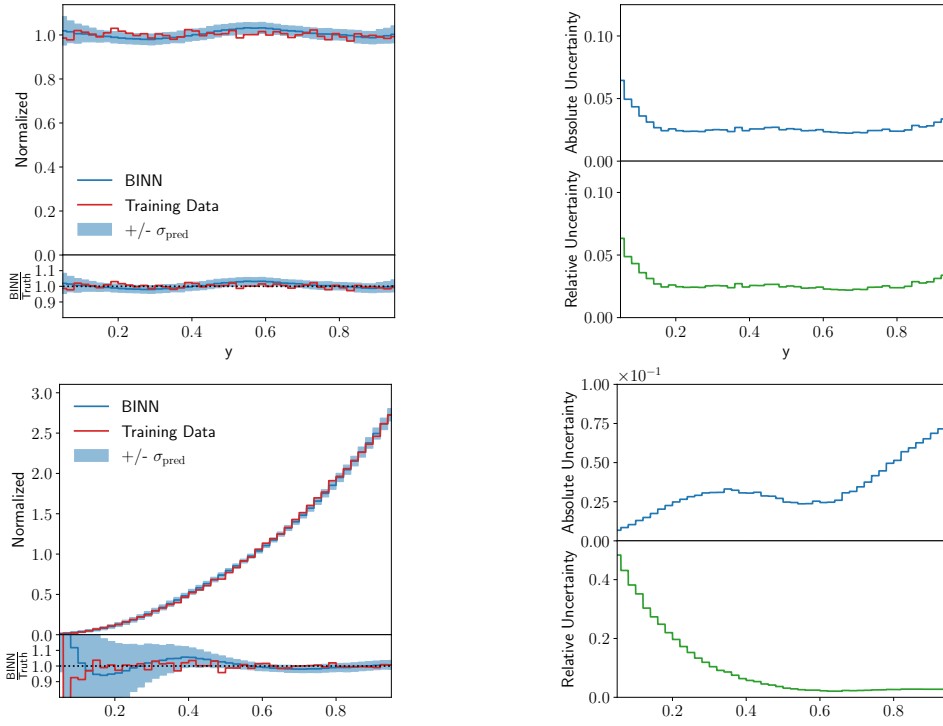

*Figure 8.* Marginalized densities and predictive uncertainties for the kicker ramp. Instead of the true distribution we now show the training data as a reference, to illustrate possible limitations. We use 10M phase space point to guarantee a stable prediction.

$\theta$. We start with one set of weights $\theta_i$ from the weight distributions, based on one random number per INN weight. We now sample $N$ points in the latent space, $z_j$, and compute $N$ phase space point $x_j$ using the BINN configuration $\theta_i$. We then bin the wanted phase space direction $x$ and approximate $p(x|\theta_i)$ by a histogram. We repeat this procedure $i = 1 ... M$ times to extract $M$ histograms with identical binning. This allows us to compute a mean and a standard deviation from $M$ histograms to approximates $p(x)$ and $\sigma_{\text{pred}}(x)$. The approximation of $\sigma_{\text{pred}}$ should be an over-estimate, because it includes the statistical uncertainty related to a finite number of samples per bin. For $N \gg 1$ this contribution should become negligible. With this procedure we effectively sample $N \times M$ points in phase space.

Following (12), we can also fix the phase space points, so instead of sampling for each weight sample another set of phase space points, we use the same phase space points for each weight sampling. This should stabilize the statistical fluctuations, but with the drawback of relying only on an effective number of $N$ phase space points. Both approaches lead to the same $\sigma_{\text{pred}}$ for sufficiently large $N$, which we typically set to $10^5 ... 10^6$. For the Bayesian weights we find stable results for $M = 30 ... 50$.

In Fig. 8 we show the marginalized densities and predictive

uncertainties for the kicker ramp. In $y$-direction the density and the predictive uncertainty show the expected flat behavior. The only exception are the phase space boundaries, where the density starts to deviate slightly from the training data and the uncertainty correctly reflects that instability. In $x$-direction, the marginalized density and uncertainty can be compared to their one-dimensional counterparts in Fig.4. While we expect the same peak-dip structure, the key question is if the numerical values for $\sigma_{\text{pred}}(x)$ change. If the network learns the $y$-direction as uncorrelated additional data, the marginalized uncertainty should decrease through a larger effective training sample. This is what we typically see for Monte Carlo simulations, where a combination of bins in an unobserved directions leads to the usual reduced statistical uncertainty. On the other hand, if the network learns that the $y$-directions is flat, then adding events in this direction will have no effect on the uncertainty of the marginalized distribution. This would correspond to a set of fully correlated bins, where a combination will not lead to any improvement in the uncertainty. In Fig. 8 we see that the $\sigma_{\text{pred}}(x)$ values on the peak, in the dip, and to the upper end of the phase space boundary hardly change from the one-dimensional results in Fig.4. This confirms our general observation, that the (B)INN learns a functional form of the density in both directions, in close analogy to a fit. It also means that the uncertainty from the generative

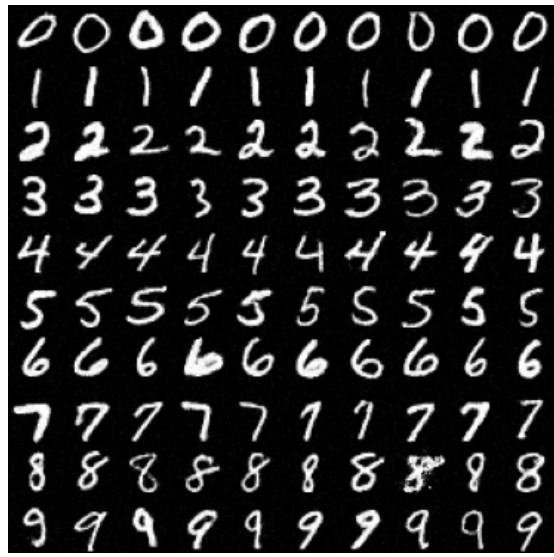

*Figure 9.* Samples of a conditional (i.e. given label information) BINN trained on the MNIST data set.

network training is not described by the simple statistical scaling as was observed for simpler networks (Bollweg et al., 2020; Kasieczka et al., 2020) and instead points towards a GANplification-like (Butter et al., 2020a) pattern.

### C.1.5. MNIST

Our low-dimensional experiments on the synthetic data sets allowed us to visualize the behaviour of the model. In order to demonstrate that the approach also scales to high-dimensional latent spaces we close this section with a small experiment on the MNIST data set. We train a conditional BINN, i.e. an BINN that gets the label information as part of the training data to allow for a nicer visualization, on the MNIST data set. Figure 9 shows ten random samples per label of the trained model, demonstrating the scalability of the BINN. We leave a proper discussion and evaluation of the learned uncertainties in the normalizing flow for further work. The conditional BINN was trained as the other models discussed in the experiments with the sole modification of an inclusion of gradient clipping which became necessary in some training runs to ensure numerical stability.

### C.2. LHC Experiment

#### C.2.1. DATA PREPARATION AND EXPLANATION

The total probability of a certain process to occur, i.e. the total probability that the collision of 2 particles with 4-momenta $p_a, p_b$ will produce $n$ particles with 4-momenta $p_1, \ldots, p_n$ is given by the cross section

$$\sigma = \int d\Phi_{2 \to n} \frac{|\mathcal{M}(p_a, p_b; p_1 \ldots p_n)|^2}{2\hat{s}} \quad (15)$$

where $\hat{s}$ is the energy of the process in the center of mass reference frame, the phase-space factor is given by

$$d\Phi_{2 \to n} = (2\pi)^4 \delta^{(4)}(p_a + p_b - p_1 \ldots - p_n) \times$$
$$\times \prod_{f=1}^{n} \frac{dp_f^3}{(2\pi)^3} \frac{1}{2p_f^0}, \quad (16)$$

and the matrix element $\mathcal{M}(p_a, p_b; p_1 \ldots p_n)$ is a quantity computed from first principles, which depends on the theoretical framework (e.g. what are the allowed interactions between particles, what is their intensity, what are the symmetries of the theory). Finally, let us recall that a particle is described in terms of its 4-momentum $p = (E, \vec{p})$, with energy and 3-momentum related by

$$p^2 = E^2 - \vec{p}^2 = m^2, \quad (17)$$

where $m^2$ is the mass of the particle.

In practice, we can think of the cross section as the normalization of a joint probability density

$$\sigma = \int dx \, p(x), \quad (18)$$

with $p(x)$ collecting both the phase-space factor and the matrix element, so that a single collision corresponds to a sample from $p(x)$. The data employed in the LHC physics section has been prepared using MADGRAPH5 (Alwall et al., 2014), a state-of-the-art software in high energy physics for sampling from $p(x)$. In practice, the data consists of a point cloud with each entry given by the set of momenta of the particle produced in the collision $(E_1, \vec{p}_1), \ldots, (E_n, \vec{p}_n)$. In the specific process under consideration, an electron-positron pair $e^- e^+$ is produced in the collision, and the data set consists of 1M lists $(E_{e-}, \vec{p}_{e-}), (E_{e+}, \vec{p}_{e+})$. The histograms displayed in Fig. 2 are some of the marginal distributions of the 4-momenta of the electron-positron pair, or of quantities directly computing in terms of 4-momenta. A comprehensive list of the quantities used is

$$E_i = \text{0-entry of 4-vector } i$$
$$p_{x,i}(p_{y,i}, p_{z,i}) = \text{1- (2-, 3-) entry of 4-vector } i$$
$$p_{T,i} = \sqrt{p_{x,i}^2 + p_{y,i}^2}$$
$$E_{i,j} = E_i + E_j$$
$$M_{i,j} = \sqrt{(E_i + E_j)^2 - (\vec{p}_i + \vec{p}_j)^2}.$$

With the exception of $E_i$ and $p_{T,i}$, which are defined in $(0, \infty)$, all other quantities are defined in the range $(-\infty, +\infty)$ in energy units.

#### C.2.2. FURTHER DISCUSSION

In Fig. 10 we show a set of absolute and relative uncertainties from the BINN. The strong peak combined with a narrow tail in the $E_e$ distribution shows two interesting features.

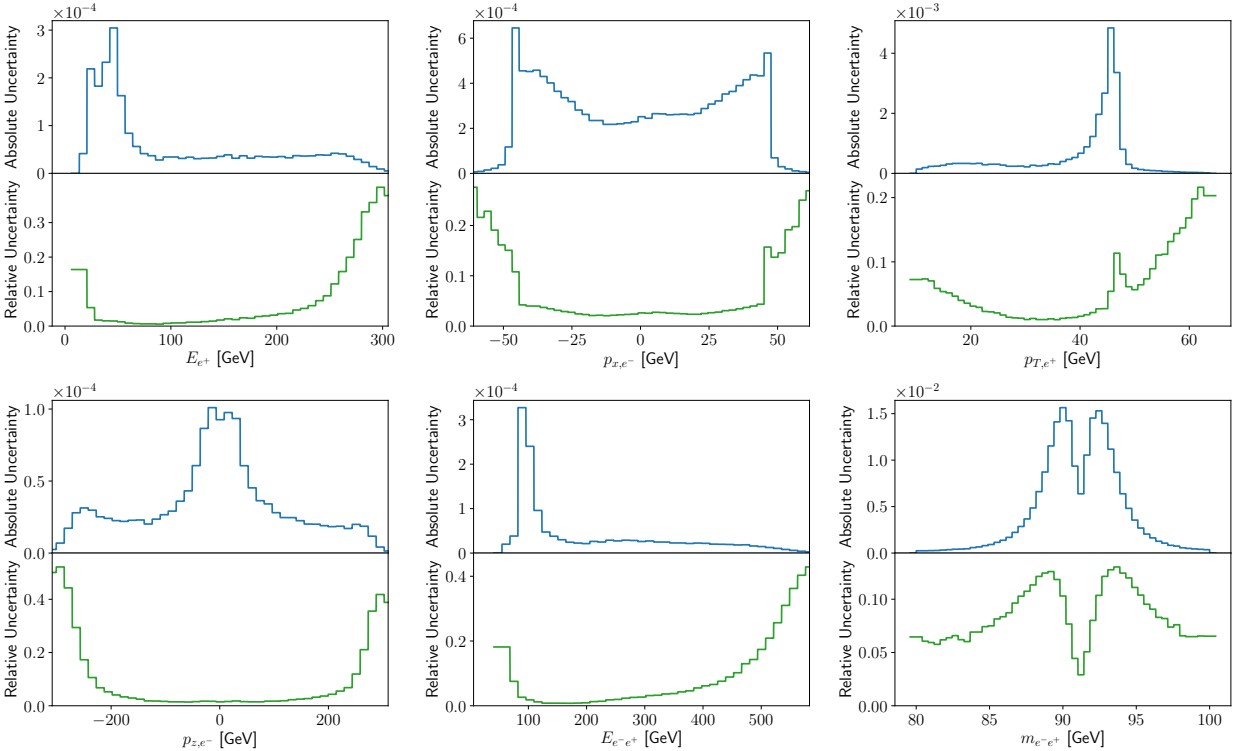

*Figure 10.* Absolute and relative uncertainties as a function of some of the kinematic Drell-Yan observables shown in Fig. 2.

Just above the peak the absolute uncertainty drops more rapidly than expected, a feature shared by the wedge and kicker ramps at their respective upper phase space boundaries. The shoulder around $E_e \approx 280$ GeV indicates that for a while the predictive uncertainty follows the increasingly poor modelling of the phase space density by the BINN, to a point where the network stops following the truth curve altogether and the predictive uncertainty is limited by the expressive power of the network. Unlike the absolute uncertainty, the relative uncertainty keeps growing for increasing values of $E_e$. This behavior illustrates that in phase space regions where the BINN starts failing altogether, we cannot trust the predictive uncertainty either, but we see a pattern in the intermediate phase space regime where the network starts failing.

The second kinematic quantity we select is the $x$-component of the momentum. It forms a relative flat central plateau with sharp cliffs at each side. Any network will have trouble learning the exact shape of such sharp phase space patterns. Here the BINN keeps track of this, the absolute and the relative predictive uncertainties indeed explode. The only difference between the two is that the (learned) density at the foot of the plateau drops even faster than the learned absolute uncertainty, so their ratio keeps growing.

Finally, we show the result for the Breit-Wigner mass peak, the physical counterpart of the Gaussian ring model of

Sec. C.1.2. Indeed, we see exactly the same pattern, namely a distinctive minimum in the predictive uncertainty right on the mass peak. This pattern can be explained by the network learning the general form of a mass peak and then adjusting the mean and the width of this peak. Learning the peak position leads to a minimum of the uncertainty right at the peak, and learning the width brings up two maxima on the shoulders of the mass peak. In combination Fig. 2 and 10 clearly show that the BINN traces uncertainties in generated LHC events just as for the toy models. Again, some distinctive patterns in the predictive uncertainty can be explained by the way the network learns the phase space density.

### C.2.3. FURTHER DETAILS

The hyperparameters and architecture for the LHC experiment are given in Table 2.

*Table 2.* Hyper-parameters for the Drell-Yan data set, implemented in PYTORCH (v1.4.0) (Paszke et al., 2019).

| Parameter | Flow |
|---|---:|
| Hidden layers (per block) | 2 |
| Units per hidden layer | 64 |
| Batch size | 512 |
| Epochs | 500 |
| Trainable weights | $\sim 182$k |
| Number of training events | $\sim 1$M |
| Optimizer | Adam |
| $(\alpha, \beta_1, \beta_2)$ | $(1 \times 10^{-3}, 0.9, 0.999)$ |
| Coupling layers | 20 |
| Prior width | 1 |