# OpenReview forum: "Understanding Event-Generation Networks via Uncertainties"
_ICML.cc/2021/Workshop/INNF — INNF+ 2021 poster_

### Official Review · Reviewer_Tr4D · 2021-06-08

**Rating:** Borderline Accept
**Confidence:** 2

**Summary:**

The paper introduces an approach to measure sampling uncertainty of flow-based generative models. It suggests doing full Bayesian inference over the flow's parameters using a variational mean-field Gaussian approximation of the parameter posterior distribution. The method is evaluated on three 2D toy problems as well as an application inspired by LHC experiments.

**Justification For Rating:**

The paper is well motivated and in most parts clearly written.

The idea of using full Bayesian inference over the flows' parameters and using a Gaussian mean-field approximation to the posterior to approximate the uncertainty of samples seems straightforward. Being no expert on Bayesian Neural Networks especially in the generative modeling domain, I cannot fully assess the novelty of the idea. I further have no intuition about scalability to higher dimensions and how much the variational gap might hamper the goal of producing accurate uncertainty estimates.

Beyond that, there are some concerns/question marks that I recommend addressing in a revision:
- RealNVP uses affine coupling layers which can be used to define diffeomorphisms on $\mathbb{R}^d$. However, two of the toy examples define densities on $[0,1]$. Using RealNVP, in this case, will result in flow densities with odd behavior on the boundary. E.g. in the ramp examples: it will be impossible to match the density on the right side. It would make sense to use some compactifying invertible layer (e.g. a sigmoid or similar) to define a proper distribution in $[0,1]$.
- The paper is not easy to understand for readers outside the domain and uses a lot of domain-specific terminologies. As this is a machine learning workshop on normalizing flows and no domain venue including a broader audience is important. Especially, the section on the LHC motivated experiment is difficult to read and understand, without knowledge about the process. I recommend briefly explain the process in simple terms to outsiders and more importantly explain the resulting random variables that make up the data set, their domain, and their dimensionality, and how it maps to the density estimation problem (e.g. what are those variables described in Fig. 2). It would help a lot to understand the merits/drawbacks of the method.
- In your introduction, you mention ensembles as an alternative to estimate uncertainty. As the shown experiments are very low dimensional: why no comparison of your model against an ensemble? This would be interesting, as your method relies on a quite simple variational approximation to the true posterior.

---

### Official Review · Reviewer_NRsw · 2021-06-11

**Rating:** Borderline Reject
**Confidence:** 4

**Summary:**

“Understanding Event-Generation Networks via Uncertainties” introduces a Bayesian invertible Neural Network (BINN) to capture uncertainty in LHC event modeling. The approach is well-motivated scientifically, and with further development could yield meaningful advances in scientific techniques for particle physics.  Uncertainty in the BINN generally followed the analytic derivations (the explanation of which I found unclear) in a toy example, and proof of concept was demonstrated in a low-dimensional Drell-Yan process.

**Justification For Rating:**

This submission appears to be the first presentation of Bayesian normalizing flows, which certainly makes it appropriate content for this workshop.  The authors should provide some introductory content on normalizing flow techniques for quantifying uncertainty in additional scientific applications, to ensure the reader that an appropriate literature search was done and this is in fact the first instance of a BINN.

Unfortunately, the LHC simulator application (which is the focus of this submission) was not explained appropriately for a non-expert audience (a machine learning conference).   There are no equations that describe the generative model in the main text (except for a zero-context process model in the first sentence of 3.2), and many variables used to describe it are introduced with no explanation.

After doing some background reading on LHC experiments, and even accounting for the excessive jargon (and grammatical mistakes), I found the description of the scientific motivation inadequate.  It failed to connect the philosophy of inference in the LHC simulator to the structure of the BINN, any more than a vague description of the different types of uncertainties that are captured in the posterior predictive distribution.  It is clear that there are different natures of uncertainty in this problem – empirical, quantum, systematic – and there are two levels of uncertainty in the BINN -- the posterior of normalizing flow weights and the stochasticity of the normalizing flow.   But, no explicit connection between these different levels of stochasticity/uncertainty and the scientifically meaningful notions of uncertainty were drawn.

Finally, normalizing flows (and this BINN setup) are perfectly capable of scaling to higher dimensions along with LHC simulator models.  This submission should have provided some scaling analysis or at least demonstrated feasibility in higher dimensions.

I would be happy to read a heavily revised, more clearly written version of this submission.

---

### Decision · Program_Chairs · 2021-06-14

**Decision:**

Accept (poster)

**Comment:**

The paper is within the scope of the workshop and considers an interesting application of Bayesian normalizing flows to high-energy physics. However, the reviewers found that the LHC application is not communicated clearly enough to non-domain-experts. We decided to accept the paper, but we urge the authors to take into account the reviewer's comments and consider how the paper can be improved so it's more accessible to non-domain-experts.